# Awareness of fifth metatarsal stress fractures among soccer coaches in Japan: A cross-sectional study

**Takayuki Miyamori**[1]*, **Masashi Aoyagi**[2], **Yu Shimasaki**[3], **Masafumi Yoshimura**[3]

**1** Faculty of Health Science, Department of Physical Therapy, Juntendo University, Bunkyo City, Tokyo, Japan, **2** Juntendo Administration for Sports, Health and Medical Sciences, Juntendo University, Hongo Bunkyo-ku, Tokyo, Japan, **3** Faculty of Health and Sports Science, Department of Health and Sports Science, Juntendo University, Inzai City, Chiba, Japan

* t.miyamori.hi@juntendo.ac.jp

**Data Availability Statement:** All relevant data are within the paper and its Supporting Information files.

**Funding:** This work was supported by a grant from the Institute of Health and Sports Science and

## Abstract

Although a fifth metatarsal stress fracture is the most frequent stress fracture in soccer players, awareness of fifth metatarsal stress fractures among soccer coaches is unclear. Therefore, we performed an online survey of soccer coaches affiliated with the Japan Football Association to assess their awareness of fifth metatarsal stress fractures. A total of 150 soccer coaches were invited for an original online survey. Data on participants' age, sex, types of coaching licence, coaching category, types of training surface, awareness of fifth metatarsal stress fractures, and measures employed to prevent fifth metatarsal stress fractures were collected using the survey. Data from 117 coaches were analysed. Eighty-seven of the 117 coaches were aware of fifth metatarsal stress fractures; however, only 30% reported awareness of preventive and treatment measures for fifth metatarsal stress fractures. Licensed coaches (i.e., licensed higher than level C) were also more likely to be aware of fifth metatarsal stress fractures than unlicensed coaches were. Furthermore, although playing on artificial turf is an established risk factor for numerous sports injuries, soccer coaches who usually trained on artificial turf were more likely to be unaware of the risks associated with fifth metatarsal stress fractures than coaches who trained on other surfaces were (e.g., clay fields). Soccer coaches in the study population were generally aware of fifth metatarsal stress fractures; however, most were unaware of specific treatment or preventive training strategies for fifth metatarsal stress fractures. Additionally, coaches who practised on artificial turf were not well educated on fifth metatarsal stress fractures. Our findings suggest the need for increased awareness of fifth metatarsal stress fractures and improved education of soccer coaches regarding injury prevention strategies.

## Introduction

A fifth metatarsal stress fracture (MT-5) is the most frequent stress fracture in soccer players. In injury surveys of European professional soccer players, the incidence of stress fractures was

Medicine, Juntendo University. (grant number 2022) The funders had no role in study design, data collection and analysis, decision to publish, or preparation of the manuscript.

**Competing interests:** The authors have declared that no competing interests exist.

approximately 5% of all injuries, 78% of which were MT-5 [1]. It frequently occurs at the zones II and III in the MT-5 Torg classification [2], which are located in the proximal shaft of the fifth metatarsal and known as Jones fractures [3]. MT-5 is mainly treated through surgery, as it is difficult to achieve bone union with conservative treatment due to the poor blood supply at the fifth metatarsal [4]. The estimated return-to-play time for soccer players with MT-5 is generally between 3 and 5 months; therefore, MT-5 is considered a career-altering obstacle [4].

Because of the high incidence and substantial return-to-play time of MT-5, team staff need to consider the risk factors for developing MT-5 during training. In previous studies, age and sex were found to be risk factors for MT-5 [5], with male athletes, in particular, being at greater risk than female athletes [6]. In addition, a deficiency in 25-hydroxyvitamin D, which is an indicator of calcium metabolism [7], foot distraction tendencies during loading [8], limited range of motion of hip internal rotation [7], and toe grip muscle weakness [8] have been reported to be associated with the occurrence of MT-5.

Doctors and physical therapists are sometimes present at the training sites during practice sessions. Additionally, they have opportunities to provide MT-5 prevention advice and training to athletes during regular training and games. However, coaches' awareness regarding MT-5 prevention is unknown. Therefore, we conducted a questionnaire survey of soccer coaches affiliated with the Japan Football Association (JFA), and the purpose of this study was to investigate their awareness of MT-5 and its prevention and treatment. We hypothesized that awareness of MT-5 among soccer coaches varies with the level of licensure and the coaching environment.

## Materials and methods

A cross-sectional study was conducted. A total of 150 soccer coaches on the JFA official team were invited to participate in this study through the JFA Official Coaching Course at Juntendo University in Japan and by direct email between the 15th of January and the 30th of March 2020. A computer-based survey was created following a literature review of the injury, treatment, and prevention of MT-5 and a discussion with experienced physical therapists, sports medicine physicians, and sports scientists who were familiar with MT-5. The final version of the questionnaire was established through consensus among all members to encompass all relevant aspects of MT-5. This study's main interest was to understand differences in the awareness about the injury, treatment, and prevention of MT-5 by licence level and coaching environment. Hence, the questionnaire was designed with an emphasis on how many coaches were aware of this information rather than the depth of their knowledge of MT-5. A research request was sent to potential respondents who provided consent. Each participant entered the mobile terminal using a personalised code. The exclusion criteria were failure to provide informed consent to complete the survey. Additionally, if more than one incomplete response in the questionnaire was confirmed, all the individual's data were excluded from the analysis. The inclusion and exclusion criteria used are indicated in Fig 1.

The survey items included age, sex, years of playing soccer, years of coaching experience, level of JFA licence (none, D, C, B, A, and S, which is the highest level of licence), coaching category (elementary student–J league), type of training surface (artificial turf and other than artificial turf), training frequency per week, training hours per session, awareness of MT-5, and awareness of treatment methods and preventive training strategies for MT-5. MT-5 prevention methods were assessed using a free-format question, while other items were assessed using multiple-choice questions. In addition, 20 participants among all responders across all coaching categories were randomly assigned to complete the survey a second time 1 week after completing their first survey to identify the reliability of the self-reported questionnaire.

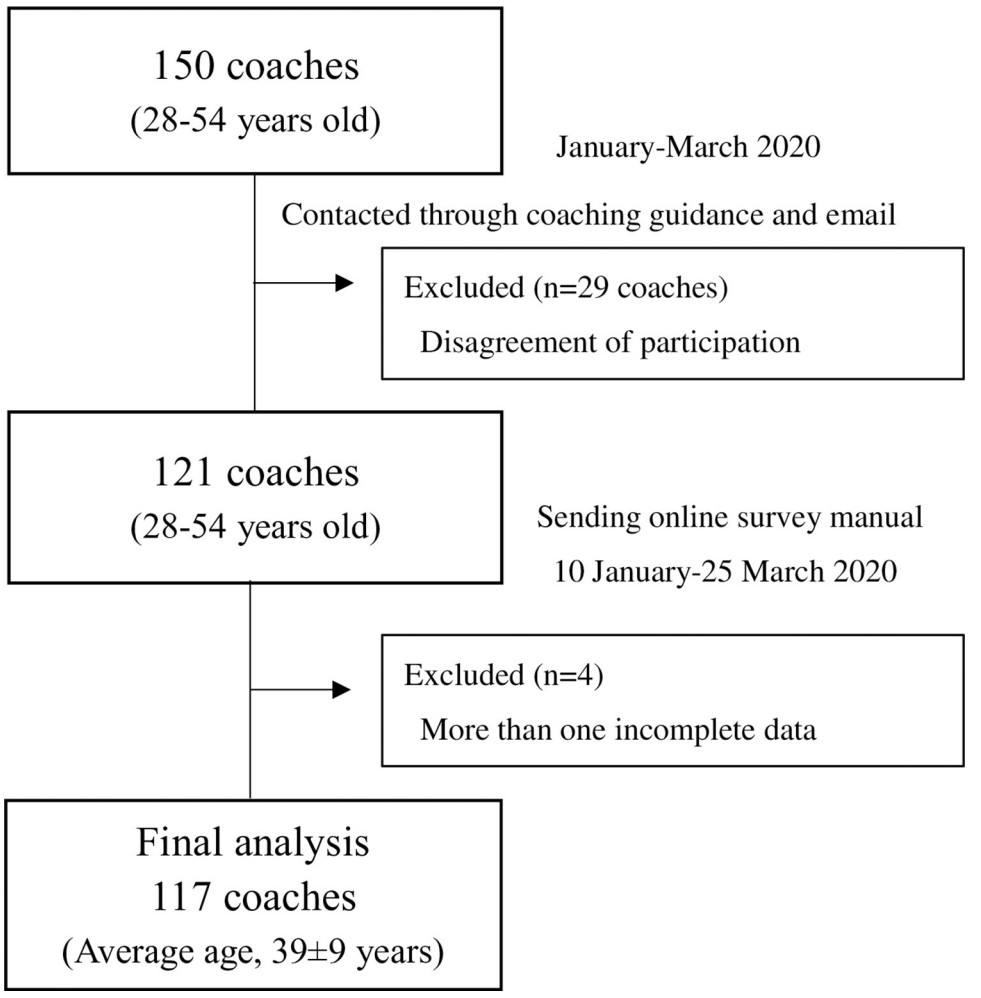

**Fig 1. Flowchart of participant recruitment and screening.**

Statistical analyses were performed using SPSS version 24 (IBM Corp., Armonk, NY). The intraclass correlation coefficients (ICC) and kappa statistics were used to measure the degree of agreement between responses from the 20 participants to confirm the test-retest reliability of the questionnaire. The number of respondents and the overall proportion of responses for each question item in the survey were calculated. Univariate analysis was performed to examine the relationship between licence type and MT-5 awareness. Additionally, the odds ratio (OR), 95% confidence interval, and $p$-value of each licence were calculated with 'no licence' as the reference value. Furthermore, a chi-square test was performed to examine the relationship between training surface types and awareness of MT-5. These analyses were performed for all responders, while the data from the first survey was employed for those who responded twice for the test-retest reliability. Statistical significance was set at $p < 0.05$.

The survey was anonymised to ensure privacy. A research ID corresponding to the participant's name was created. Registration for the survey was also conducted using the research ID. An encrypted passcode was set separately from the research ID to prevent unauthorised access, and responses could only be made by entering this passcode to ensure confidentiality. This study was approved by the International University of Health and Welfare Ethics Review Board (approval number: 16-Io-203). All survey respondents provided written informed consent.

## Results

We contacted 150 potential participants, and the data from 117 male respondents (average age: 39 ± 9 years, effective response rate: 78%) were analysed. The ICC and kappa values for the test-retest reliability were excellent (ICC range = 0.88–0.98) and moderate to almost perfect (k range = 0.57–0.88), respectively. Table 1 shows the number and proportion of responses for each question item in the survey. The coaching category with the highest frequency was high school, which accounted for 51.3% of the study population. More than half (54.7%) of the respondents coached their teams on artificial turf. Approximately 30% of respondents had an A-level licence, with the fewest respondents having a D-level licence. Of the 117 total respondents, 87 (74.4%) were aware of MT-5, although only 30% implemented treatment methods and preventive training strategies. Regarding MT-5 prevention methods, the most frequent response was adjusting the training volume; other responses included trunk training, ankle tube training, and changing spike cleats. Table 2 shows the types of licences and the results of the univariate analysis of the degree of MT-5 awareness. The univariate analysis showed that the OR of being unaware of MT-5 was lower for all licences than it was for unlicensed coaches. Holding a D-level licence reduced the odds of being unaware of MT-5 by 75%; however, this finding was not statistically significant ($p$ = 0.30). Coaches with C-level, B-level, A-level, and S-level licences were significantly more likely to be aware of MT-5 than were unlicensed coaches (all $p < 0.05$). Regarding the treatment methods, 20.0% (3/15) of the unlicensed coaches, 0% (0/3) of those in D-level, 25.7% (9/25) in C-level, 20.7% (6/29) in B-level, 34.2% (13/38) in A-level, and 57.1% (4/7) in S-level were aware of the treatment. The respondents aware of the preventive training strategies were as follows: 26.7% (4/15) of the unlicensed coaches, 0% (0/3) in D-level, 28.0% (7/25) in C-level, 27.6% (8/29) in B-level, 34.2% (13/38) in A-level, and 57.1% (4/7) in S-level. Table 3 summarises the results of the chi-square test for the type of training surface and the degree of MT-5 awareness. Coaches training on artificial turf were more likely to be unaware of MT-5 than were coaches training on other surface types ($p < 0.05$).

**Table 1. Responses to the questionnaire.**

|  | Possible responses | % | n |
|---|---|---|---|
| **Coaching categories** | Primary school | 7.7 | 9 |
|  | Junior high school | 17.1 | 20 |
|  | High school | 51.3 | 60 |
|  | University | 19.7 | 23 |
|  | Semi-professional | 3.4 | 4 |
|  | Professional | 0.9 | 1 |
| **Use of artificial turf** | Yes | 54.7 | 64 |
| **Certified licence (level)** | No licence | 12.8 | 15 |
|  | D | 2.6 | 3 |
|  | C | 21.4 | 25 |
|  | B | 24.8 | 29 |
|  | A | 32.5 | 38 |
|  | S | 6 | 7 |
| **Awareness of MT-5** | Yes | 74.4 | 87 |
| **Awareness of MT-5 treatment methods** | Yes | 29.9 | 35 |
| **Awareness of MT-5 preventive training** | Yes | 30.8 | 36 |

MT-5, fifth metatarsal stress fracture; %, percentage; n, number of coaches.

**Table 2. Relationship between licence level and MT-5 awareness.**

|  |  | OR | 95% CI | *p*-value |
|---|---|---|---|---|
| **Certified licence (level)** | ref. = no licence | 0.35 |  | <0.05 |
|  | D | 0.25 | 0.02–3.47 | 0.30 |
|  | C | 0.1 | 0.02–0.43 | <0.05 |
|  | B | 0.16 | 0.04–0.63 | <0.05 |
|  | A | 0.11 | 0.03–0.44 | <0.05 |
|  | S | 0.08 | 0.01–0.90 | <0.05 |

OR, odds ratio; 95% CI, 95% confidence interval; ref., reference.

**Table 3. Relationship between type of training surface and MT-5 awareness.**

|  |  | Type of training surface* | | Total |
|---|---|---|---|---|
|  |  | Artificial turf | Other than artificial turf |  |
| **MT-5 awareness** | Yes | 39 (60.9%) | 48 (90.6%) | 87 |
|  | No | 25 (39.1%) | 5 (9.4%) | 30 |
| **Total** |  | 64 | 53 | 117 |

*Percentage of coaches.

## Discussion

This was the first online questionnaire survey of soccer coaches affiliated with a JFA-certified team in which the licence status of coaches, training surfaces, and MT-5 awareness were examined. Among the 117 participants surveyed, 87 (74.4%) soccer coaches were familiar with MT-5. Licensed coaches (C, B, A, and S levels) were significantly more likely to be aware of MT-5 than unlicensed coaches were. However, only 30% of the coaches implemented treatment and preventive training, although the higher the class of licence, the greater the awareness of treatment and prevention strategies. Clearly, soccer coaches lack awareness of how to prevent and treat MT-5.

A previous study surveying lower extremity joint injuries among female soccer players and instructors found that, although coaches were more aware of injuries than athletes were, they did not adequately understand the risk factors for injuries or specific prevention strategies [9]. In addition to tactical and technical training instructions, the JFA official coach licence curriculum covers sports injury prevention and first aid. Furthermore, the JFA published a guide to soccer medicine for athletes and coaches in 2005 and advocated a method for evaluating the physical strength and conditioning necessary for soccer, as well as a method for managing typical orthopaedic and medical diseases [10]. Lower limb and trunk injuries account for 90% of injuries in soccer, with particular attention given to Osgood–Schlatter disease, knee and ankle joint soft tissue injuries (meniscal and ligament injuries), and lumbar injuries during the growth period in athletes [10]. However, currently, coaches do not adequately address factors related to MT-5 using preventive interventions and rehabilitation despite the significant time loss from playing soccer associated with this injury [11]. Approximately 50% of the respondents in the current study coached high school athletes; meanwhile, 54.7% of all coaches trained their athletes on artificial turf. Those who coached on artificial turf had lower awareness of MT-5 injuries than those who coached on other surfaces. Since 2001, the use of long-pile artificial turf has increased rapidly in Japan [12]. Although artificial turf has many

advantages, including reduced ground maintenance costs, epidemiological studies have shown that artificial turf causes an increased burden on the feet and knee joints, leading to greater injuries than other surfaces [13–15]. Ekstrand et al. [15] determined that training and playing matches are more frequently conducted on artificial grass than on natural grass in high school and college. Turning motions on artificial grass have been shown to cause rotational stress on the foot, and the accumulation of stress increases as the intensity of competition increases from high school to university. [14] In recent studies, players with a high body mass index also reported an increase in plantar load due to a decreased balance ability during play [16] and an increase in mechanical stress on the fifth metatarsal secondary to changes in lateral plantar pressure during turning [17]. Furthermore, Miyamori et al. [18] showed that training or playing on artificial turf increases the risk of developing MT-5 compared with clay fields. Together, these studies showed that increased play on artificial turf might increase the risk of stress fractures of the lower extremities, such as MT-5.

To reduce the incidence of MT-5, doctors and physical therapists need to provide specialized and comprehensive programs regarding medical science and preventive training methods for coaches, given the multifactorial nature of the causes of MT-5 [7,8,16–19]. Future research should focus on determining other risk factors for lower extremity stress fractures, like MT-5, which tend to be specific to soccer players. It is necessary to develop a training program that considers the risk factors for MT-5, which would improve athletes' self-management ability, including injury prevention. These approaches can strengthen individual athletes and teams.

This study had some limitations and future directions that warrant discussion. First, our sample size was relatively small; however, we could describe different aspects across a range of age groups and categories. Second, the study's questionnaire was designed to investigate the awareness of MT-5, not the depth of respondents' understanding of MT-5. Therefore, there may be differences in the level of understanding even among respondents aware of it. Third, the content validity of the questionnaire was not confirmed through qualitative research, as suggested by a previous study [20]; however, we developed it based on literature review and expert opinions. Fourth, some coaches who participated in our survey may have been medically trained to be aware of MT-5 and provide education regarding MT-5 prevention and therapy. This would relieve the pressure on the coaching staff by directing their efforts toward MT-5 awareness. Future studies must examine the composition of team staff to determine the status of the professional injury prevention system and improve athletes' awareness of sports injury management. Finally, this study was limited to the Japanese soccer community and must be expanded to include coaches and medical staff across Asia, Europe, and worldwide. Sports injury surveillance systems are currently being developed for soccer [21]. In the future, it will be necessary to integrate and disseminate all injury reports from Japanese soccer in collaboration with the JFA Sports Medicine Committee and the Japan Professional Soccer League. This approach will make it possible to identify and share risk factors for sports injuries, including MT-5, which is a universal problem, ultimately contributing to sports injury prevention.

## Conclusions

We conducted an MT-5 awareness survey among soccer coaches. Coaches without a JFA-certified coaching licence had lower awareness of MT-5 than did coaches with such licences (C level and above). In contrast, although licensed coaches understood the nature of the injury, only a third understood and implemented treatment methods and preventive training. Additionally, even though playing on artificial turf is associated with a higher injury incidence, those who coached on artificial turf had lower awareness of MT-5 than those who coached on other surfaces. In the future, JFA coach training sessions should increase coaches' awareness of

MT-5 pathologies and suggest concrete preventive training methods to reduce the risk of developing MT-5.

## Supporting information

**S1 Dataset.**
(XLSX)

## Acknowledgments

We would like to thank Ryuich Sawa and Masahiro Imagawa for their valuable support.

## Author Contributions

**Conceptualization:** Takayuki Miyamori.

**Project administration:** Yu Shimasaki.

**Supervision:** Masafumi Yoshimura.

**Writing – original draft:** Takayuki Miyamori.

**Writing – review & editing:** Masashi Aoyagi, Masafumi Yoshimura.

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
