## [Decision Letter · Decision Letter 0]

21 Feb 2023

PONE-D-22-27267Awareness of fifth metatarsal stress fractures among soccer coaches in Japan: A cross-sectional studyPLOS ONE

Dear Dr. Miyamori,

Thank you for submitting your manuscript to PLOS ONE. After careful consideration, we feel that it has merit but does not fully meet PLOS ONE’s publication criteria as it currently stands. Therefore, we invite you to submit a revised version of the manuscript that addresses the points raised during the review process.

ACADEMIC EDITOR:Dear Author,1. Please attend to all the reviewer's comments and make the necessary changes.2. Could you provide a few names of potential reviewers for the next round of review. The decision of this manuscript is justified based on PLOS ONE’s publication criteria and not on its novelty or perceived impact.

We look forward to receiving your revised manuscript.

Kind regards,

Zulkarnain Jaafar

Academic Editor

PLOS ONE

Reviewers' comments:

Reviewer's Responses to Questions

**Comments to the Author**

1. Is the manuscript technically sound, and do the data support the conclusions?

Reviewer #1: Partly

2. Has the statistical analysis been performed appropriately and rigorously? 

Reviewer #1: Yes

3. Have the authors made all data underlying the findings in their manuscript fully available?

Reviewer #1: No

4. Is the manuscript presented in an intelligible fashion and written in standard English?

Reviewer #1: Yes

5. Review Comments to the Author

Reviewer #1: I would recommend few corrections or some of my doubts need comments before the final publication:

1. The authors wrote that they examined the knowledge, however they use the word "awarness" all the time. I do not agree that those words can be used interchangably. So, what was evaluated?

2. How was the questionnaire prepared, the answers....to check knowledge? It is ood to use Likert scale or in some other form of raiting....e.g. to evaluate it. Other way, it can be only said how many people answer on sth. but how deep they knew the meaning of the question? It also influance their awarness....

3. In the Abstract the sample size os 117, but in Methods is still 150. Please, correct.

4. Some references are quite old...Ref.1

6. PLOS authors have the option to publish the peer review history of their article (what does this mean?). If published, this will include your full peer review and any attached files.

Reviewer #1: No

---

## [Author Response · Author response to Decision Letter 0]

18 May 2023

Reviewer #1: I would recommend few corrections or some of my doubts need comments before the final publication:

Thank you for your comments. We have answered them below. 

1. The authors wrote that they examined the knowledge, however they use the word "awarness" all the time. I do not agree that those words can be used interchangably. So, what was evaluated?

⇒Thank you for this question. Considering the study’s purpose and the questionnaire utilised, we believe “awareness” is a more suitable term. Therefore, we changed the term “knowledge” to “awareness” throughout the revised manuscript.

2. How was the questionnaire prepared, the answers....to check knowledge? It is ood to use Likert scale or in some other form of raiting....e.g. to evaluate it. Other way, it can be only said how many people answer on sth. but how deep they knew the meaning of the question? It also influance their awarness....

⇒Thank you for this question. We have added some comments in Materials and Methods to explain how the questionnaire was developed, as follows: “A computer-based survey was created following a literature review of the injury, treatment, and prevention of MT-5 and a discussion with experienced physical therapists, sports doctors, and sports scientists who were familiar with MT-5. This study’s main interest was to understand differences in the awareness about the injury, treatment, and prevention of MT-5 by license level and coaching environment. Hence, the questionnaire was designed with an emphasis on how many coaches were aware of this information rather than the depth of their knowledge of MT-5.”(p4 lines 73–79).

In addition, we realise that this was a limitation of the study. Hence, we have added some comments on the study’s limitations as follows: “Second, the study’s questionnaire was designed to investigate the awareness of MT-5, not the depth of respondents’ understanding of MT-5. Therefore, there may be differences in the level of understanding even among respondents aware of it.” (p10, lines 192–194)

3. In the Abstract the sample size os 117, but in Methods is still 150. Please, correct.

⇒Thank you for pointing this out. We have revised the abstract as follows: “A total of 150 soccer coaches were invited for an original online survey” (p2 lines 21–22) and “Data from 117 coaches were analysed.” (p2 line 25)

4. Some references are quite old...Ref.1

⇒We have retained reference 1 in the revised manuscript because it defined the “Torg classification.” However, we have changed the place of the reference number in the manuscript as follows: “In particular, zones II and III in the MT-5 Torg classification [1]…” (p3 line 43)

Reference 5 (Kavanaugh 1978) has been deleted from the manuscript since reference 5 (Kane 2015) can suffice. (p3 line 52).

＊In addition, we thought it would be better to add the test-retest reliability results. Therefore, we have included an additional comment on the statistical analysis as follows: “The intraclass correlation coefficients (ICC) and kappa statistics were used to measure the degree of agreement between responses from 20 participants to confirm the test-retest reliability of the questionnaire” (p5 lines 95–98) We have also revised the results section as follows: “The ICC and kappa values for the test-retest reliability were excellent (ICC range = 0.88 - 0.98) and moderate to almost perfect (k range = 0.57–0.88), respectively.” (p6 lines 115–117)

---

## [Decision Letter · Decision Letter 1]

23 Jun 2023

PONE-D-22-27267R1Awareness of fifth metatarsal stress fractures among soccer coaches in Japan: A cross-sectional studyPLOS ONE

Dear Dr. Miyamori,

Thank you for submitting your manuscript to PLOS ONE. After careful consideration, we feel that it has merit but does not fully meet PLOS ONE’s publication criteria as it currently stands. Therefore, we invite you to submit a revised version of the manuscript that addresses the points raised during the review process.

ACADEMIC EDITOR:Dear Author,Please make the necessary corrections based on the reviewer's comments. The decision of this manuscript is justified based on PLOS ONE’s publication criteria and not on its novelty or perceived impact.

We look forward to receiving your revised manuscript.

Kind regards,

Zulkarnain Jaafar

Academic Editor

PLOS ONE

Reviewers' comments:

Reviewer's Responses to Questions

**Comments to the Author**

1. If the authors have adequately addressed your comments raised in a previous round of review and you feel that this manuscript is now acceptable for publication, you may indicate that here to bypass the “Comments to the Author” section, enter your conflict of interest statement in the “Confidential to Editor” section, and submit your "Accept" recommendation.

Reviewer #2: (No Response)

2. Is the manuscript technically sound, and do the data support the conclusions?

Reviewer #2: Partly

3. Has the statistical analysis been performed appropriately and rigorously? 

Reviewer #2: Yes

4. Have the authors made all data underlying the findings in their manuscript fully available?

Reviewer #2: No

5. Is the manuscript presented in an intelligible fashion and written in standard English?

Reviewer #2: Yes

6. Review Comments to the Author

Reviewer #2: Abstract:

Line36-38: Suggest to rephrase the sentence- it was unclear

Introduction

Line 43-45: Suggest to rephrase the sentence to show that the common fracture is at the Zone II and III of the 5th MTB which also known as Jones fracture.

Line 46: To confirm reinjury is one of the factors for surgery?

Line 54: immobilization of the lateral plantar- pathological or by brace?

Line 55-56: suggest to rephrase sentence

Line 64-66: is the objective of the study?

Materials & Methods

The methodology was not very clear.

A new questionnaire was created. Validation of the questionnaire was based on the expert committee?

Test-retest reliability was done among the coaches?

Coaches for test-retest was also involved in the main study?

Was the study applied for ethics clearance?

Result

The result did not answer the objective of the study which want to study the coaches awareness of MT-5, its prevention and treatment.

In the result- the correlation was described in detail; however, the prevention and treatment strategies were covered superficially.

Discussion

Good discussion, but suggest to include other related studies that involved 5th MTB stress fracture

Suggest to provide the sample of the online survey as supplement

5MT- not sure if it is acceptable as acronym

Too many semicolon (;)

May need to check the writing style

7. PLOS authors have the option to publish the peer review history of their article (what does this mean?). If published, this will include your full peer review and any attached files.

Reviewer #2: No

---

## [Author Response · Author response to Decision Letter 1]

30 Aug 2023

We thank the Editor and the reviewers for their thoughtful suggestions and insights. The manuscript has benefited from these insightful suggestions. 

The manuscript has been rechecked and the necessary changes have been made in accordance with the reviewers’ suggestions. The revisions incorporated are indicated via tracked changes. The responses to all comments have been prepared in a point-by-point manner and attached herewith. We hope that the responses have addressed all the reviewers’ concerns satisfactorily.

---

## [Decision Letter · Decision Letter 2]

20 Sep 2023

PONE-D-22-27267R2Awareness of fifth metatarsal stress fractures among soccer coaches in Japan: A cross-sectional studyPLOS ONE

Dear Dr. Miyamori,

Thank you for submitting your manuscript to PLOS ONE. After careful consideration, we feel that it has merit but does not fully meet PLOS ONE’s publication criteria as it currently stands. Therefore, we invite you to submit a revised version of the manuscript that addresses the points raised during the review process.

ACADEMIC EDITOR: Dear Author,Please attend to all of the reviewer's comments and make the necessary changes. The decision  of this manuscript is justified based on PLOS ONE’s publication criteria and not on its novelty or perceived impact.

We look forward to receiving your revised manuscript.

Kind regards,

Zulkarnain Jaafar

Academic Editor

PLOS ONE

Reviewers' comments:

Reviewer's Responses to Questions

**Comments to the Author**

1. If the authors have adequately addressed your comments raised in a previous round of review and you feel that this manuscript is now acceptable for publication, you may indicate that here to bypass the “Comments to the Author” section, enter your conflict of interest statement in the “Confidential to Editor” section, and submit your "Accept" recommendation.

Reviewer #3: All comments have been addressed

Reviewer #4: (No Response)

2. Is the manuscript technically sound, and do the data support the conclusions?

Reviewer #3: Yes

Reviewer #4: No

3. Has the statistical analysis been performed appropriately and rigorously? 

Reviewer #3: I Don't Know

Reviewer #4: Yes

4. Have the authors made all data underlying the findings in their manuscript fully available?

Reviewer #3: No

Reviewer #4: Yes

5. Is the manuscript presented in an intelligible fashion and written in standard English?

Reviewer #3: Yes

Reviewer #4: Yes

6. Review Comments to the Author

Reviewer #3: Dear authors,

I found this version a completely revised version of your paper based on all previous authors' comments.

I think you have addressed all and I recommend acceptance.

Reviewer #4: The article highlights important subject. The goals of this work are a reminder for all coaches and researchers to pay attention to the coaches’ awareness of MT-5, its prevention and treatment.

After reading the entire manuscript, I was disappointed that the methodology was not very strong. There is uncertainty about important things.

Major revision

Material and method

• The authors have missed something crucial. The study fails to address content validity. Content validity is a crucial aspect of testing or measuring anything.

Minor revision

Introduction

Good introduction!

• lines 51-53: Please move the sentence to line 4 to emphasize the incidence.

• The prevalence of MT-5 is different in non-athletic men and women and athletes. Boutefnouchet et al (https://doi.org/10.1177/1460408614525738 ). Suggest to add information about gender differences in the introduction.

Material and method

• Line 80: What do the authors mean by sports doctors? If the authors mean sports medicine physician, suggest to rephrase it.

• Lines 93-97: Since one of the objectives of this article was to investigate awareness about MT-5 therapy, the authors neglected to mention it.

• Please clarify how the authors assessed MT-5 treatment awareness in the method section.

Results

• One of the characteristics of the participants was their gender. The authors did not mention any results about it. Please add information about gender distribution.

• Lines 130-131: “MT-5 prevention methods were assessed using a free-format question”. Suggest to move this sentence to the material and method section.

• Table 1: Last row “preventive training”: the objective was to investigate the awareness of prevention, suggest to rewrite it.

• Table 2 and table 3: The authors used “MT-5 recognition” in the title. Suggest to rephrase it to “MT-5 Awareness”.

• Please provide the sample of the online survey as supplement.

7. PLOS authors have the option to publish the peer review history of their article (what does this mean?). If published, this will include your full peer review and any attached files.

Reviewer #3: No

Reviewer #4: No

---

## [Author Response · Author response to Decision Letter 2]

11 Jan 2024

Comments to the Author

1. If the authors have adequately addressed your comments raised in a previous round of review and you feel that this manuscript is now acceptable for publication, you may indicate that here to bypass the “Comments to the Author” section, enter your conflict of interest statement in the “Confidential to Editor” section, and submit your "Accept" recommendation.

Reviewer #3: All comments have been addressed

Reviewer #4: (No Response)

2. Is the manuscript technically sound, and do the data support the conclusions?

Reviewer #3: Yes

Reviewer #4: No

3. Has the statistical analysis been performed appropriately and rigorously?

Reviewer #3: I Don't Know

Reviewer #4: Yes

4. Have the authors made all data underlying the findings in their manuscript fully available?

Reviewer #3: No

Reviewer #4: Yes

5. Is the manuscript presented in an intelligible fashion and written in standard English?

Reviewer #3: Yes

Reviewer #4: Yes

6. Review Comments to the Author

Reviewer #3: Dear authors,

I found this version a completely revised version of your paper based on all previous authors' comments.

I think you have addressed all and I recommend acceptance.

Reviewer #4: The article highlights important subject. The goals of this work are a reminder for all coaches and researchers to pay attention to the coaches’ awareness of MT-5, its prevention and treatment.

After reading the entire manuscript, I was disappointed that the methodology was not very strong. There is uncertainty about important things.

Major revision

Material and method

• The authors have missed something crucial. The study fails to address content validity. Content validity is a crucial aspect of testing or measuring anything.

Minor revision

Introduction

Good introduction!

• lines 51-53: Please move the sentence to line 4 to emphasize the incidence.

• The prevalence of MT-5 is different in non-athletic men and women and athletes. Boutefnouchet et al (https://doi.org/10.1177/1460408614525738 ). Suggest to add information about gender differences in the introduction.

Material and method

• Line 80: What do the authors mean by sports doctors? If the authors mean sports medicine physician, suggest to rephrase it.

• Lines 93-97: Since one of the objectives of this article was to investigate awareness about MT-5 therapy, the authors neglected to mention it.

• Please clarify how the authors assessed MT-5 treatment awareness in the method section.

Results

• One of the characteristics of the participants was their gender. The authors did not mention any results about it. Please add information about gender distribution.

• Lines 130-131: “MT-5 prevention methods were assessed using a free-format question”. Suggest to move this sentence to the material and method section.

• Table 1: Last row “preventive training”: the objective was to investigate the awareness of prevention, suggest to rewrite it.

• Table 2 and table 3: The authors used “MT-5 recognition” in the title. Suggest to rephrase it to “MT-5 Awareness”.

• Please provide the sample of the online survey as supplement.

7. PLOS authors have the option to publish the peer review history of their article (what does this mean?). If published, this will include your full peer review and any attached files.

Do you want your identity to be public for this peer review? For information about this choice, including consent withdrawal, please see our Privacy Policy.

Reviewer #3: No

Reviewer #4: No

---

## [Editor Report · Decision Letter 3]

17 Jan 2024

Awareness of fifth metatarsal stress fractures among soccer coaches in Japan: A cross-sectional study

PONE-D-22-27267R3

Dear Dr. Miyamori,

We’re pleased to inform you that your manuscript has been judged scientifically suitable for publication and will be formally accepted for publication once it meets all outstanding technical requirements.

Kind regards,

Zulkarnain Jaafar

Academic Editor

PLOS ONE
---

## [Editor Report · Acceptance letter]

30 Apr 2024

PONE-D-22-27267R3 

PLOS ONE

Dear Dr. Miyamori, 

I'm pleased to inform you that your manuscript has been deemed suitable for publication in PLOS ONE. Congratulations! Your manuscript is now being handed over to our production team.

Kind regards, 

on behalf of

Dr. Zulkarnain Jaafar 

Academic Editor

PLOS ONE